# Differential Kalman Filter Design for GNSS Open Loop Tracking

**Tian Jin [1,2]** [ORCID]**, Heliang Yuan [1]** [ORCID]**, Keck-Voon Ling [3], Honglei Qin [1,*] and Jianrong Kang [2,4]**

[1] School of Electronic and Information Engineering, Beihang University, 37 Xueyuan Road, Haidian District, Beijing 100191, China; jintian@buaa.edu.cn (T.J.); yuanheliang@buaa.edu.cn (H.Y.)

[2] Shaanxi Key Laboratory of Integrated and Intelligent Navigation, Xi'an 710068, China; 13772138621@163.com

[3] School of Electrical and Electronic Engineering, Nanyang Technological University, 50 Nanyang Avenue, Singapore 639798, Singapore; ekvling@ntu.edu.sg

[4] Xi'an Research Institute of Navigation Technology, Xi'an 710068, China

* Correspondence: ateqhl@buaa.edu.cn

**Abstract:** Global navigation satellite system (GNSS) positioning in an urban environment is in need for accurate, reliable and robust positioning. Unfortunately, conventional closed-loop tracking fails to meet the demand. The open loop tracking shows improved robustness, however, the precision is unsatisfactory. We propose a differential Kalman filter for open loop, of which the measurement vector contains the differential values of open loop navigation results between adjacent epochs. The differential Kalman filter makes use of the satellite geometry (i.e., spatial domain) and motion relationship (i.e., temporal domain) to filter frequency and code phase estimations of conventional open loop tracking. The improved performances of this architecture have been analyzed theoretically and demonstrated by road tests in an urban environment. The proposed architecture shows more than 50% accuracy improvement than the conventional open-loop tracking architecture.

**Keywords:** GNSS; open loop; differential Kalman filter

## 1. Introduction

Global navigation satellite system (GNSS) has been widely used over the past several decades, and demand for GNSS receivers to operate in a challenging environment is increasing. The traditional GNSS receivers usually employ 8–12 scalar tracking loops, processing each channel independently. However, the performances of the traditional receiver will deteriorate in low signal-to-noise ratio (SNR) or high dynamic environments, and it will even completely lose lock in the worst case.

To improve the reliability and robustness of scalar tracking loops, some optimization methods are proposed. Curran [1] analyzed design and performance of discrete-time frequency-locked loop (FLL). Unambiguous frequency aided (UFA) phase-locked loop (PLL) was presented to combine frequency and phase tracking together without ambiguity [2]. Optimized carrier tracking loop design for real-time high-dynamics GNSS receivers was introduced [3]. Yang [4,5] introduced a generalized theoretical and optimal framework for PLL and FLL. On the other hand, a Kalman filter (KF) is introduced to scalar tracking loops to substitute conventional filter. Psiaki [6] analyzed the performances of the KF in a weak signal, while Ziedan [7] offered a more comprehensive analysis. Omidi et al. [8] and Gazor et al. [9] analyzed the structure and performance of the differential Kalman filter. Another method to improve position accuracy is differential position. Zhao [10] introduced using KF to estimate ambiguity in RTK under multi-constellation condition, enhancing positioning precision and availability.

A closed-loop architecture with negative feedback could easily lose lock in challenging environments. To improve the robustness, an open loop tracking (OL) [11] that precisely acquires a

signal parameter as the tracking result is presented. The feed-forward estimation technique employed by open loop tracking can effectively overcome the limitation of traditional closed-loop sequential architecture [12]. At the same time, the open loop combines acquisition and tracking together, making it possible to track signal in dynamic environment [13]. An open loop tracking aided by the Kalman filter (OL-KF) achieves a better tracking accuracy performance [14]. Han et al. [15] combined open loop with an unscented Kalman filter (UKF) for high dynamic carrier tracking. Tahir et al. [16] combined open loop with smooth filters for carrier recovery. These architectures were named "quasi-open-loop".

The pseudorange and pseudorange rate estimated by open loop tracking approaches are affected by different kinds of propagation errors. Considering these common errors can be eliminated by differential methods between adjacent epochs, we propose an open loop with a differential Kalman filter (OL-DKF). The proposed architecture focuses on combining the tracking result from all channels as input and the position solution as output.

First, the architecture of OL-DKF is introduced. Second, the performance of OL-DKF is analyzed theoretically. Third, theoretical performance comparison among open loop, open loop with Kalman filter and the proposed open loop with a differential Kalman filter is presented. Finally, data collected from road tests in different urban test cases are used to demonstrate improvement of the proposed architecture.

## 2. Structure of OL-DKF

The architecture of the open loop (OL) is presented in Figure 1. First, an initial position, velocity and time (PVT) solution is obtained from the conventional receiver. Second, a two-step procedure is carried out to obtain a navigation solution. The code phase and Doppler frequency are first estimated from a two-dimensional correlation function for each tracking channel. The second step is that pseudoranges and pseudorange rates are calculated by code phases and frequencies, assisted with the initialization results. Finally, the calculation of the position, velocity and time (PVT) is performed. This approach does not exploit the geometrical relationships between the satellites and the receiver, as well as the motion relationships between different epochs.

To improve the accuracy of open loop tracking, the architecture of OL-KF is proposed in Figure 2 [14]. The initial receiver PVT is also obtained at first. Then, a two-step procedure is also carried out to obtain the subsequent PVT solutions. In the first step, the code phase and Doppler frequency are obtained in each channel, which is similar to the first step of the open loop. Then, the Doppler frequency is filtered by the Kalman filter and the code phase is smoothed by the filtered Doppler frequency. In the second step, pseudoranges and pseudorange rates are obtained to calculate PVT, which is also similar to second step of open loop. The tracking accuracy of this model is higher than that of open loop because the model exploits the motion relationship between pseudorange and pseudorange rate of each channel in filter procedure. However, it does not take account of the geometrical relationships because the Kalman filter in each channel is independent with the others.

To utilize the geometrical relationship, a new architecture, open loop with a differential Kalman filter, was proposed and shown in Figure 3. The receiver makes an initial PVT solution based on the conventional receiver architecture. Then, a two-step procedure is carried out. In the first step, the differential values of the code phase and Doppler frequency between adjacent epochs for all tracking channels are obtained. The benefits of using the differential values include (a) exploiting temporal correlation of tracking result and (b) mitigating common propagation errors in the estimations of code phase and Doppler frequency. In the second step, the differential values are used as the measurement for the differential Kalman filter to compute the navigation solution. The advantages of the proposed approach are: (a) All channels are combined by the differential Kalman filter. (b)'The differential position and velocity as state vector exploit the motion relationship. (c) OL-KF filters independently tracking result of each channel and separates tracking filter and position procedures. However, OL-DKF utilizes the geometrical relationships between the receiver and the satellites to filter jointly all the tracking results with one filter and combines filter with position procedures. (d) Different from

using pseudoranges and pseudorange rates to calculate PVT in OL and OL-KF, the OL-DKF utilizes differential code phases and Doppler frequencies to position. Thus, the OL-DKF can achieve better positioning accuracy. In the OL search step, different correlation time can be selected according to the different carrier-to-noise ratio of signal [17]. The acquisition search will be carried out in the bins near the predicted values at the previous epoch. It will shorten the search dwell time. In Figures, it should be noted that $\left[\tau_k^1, \ldots, \tau_k^N\right]$ and $\left[f_k^1, \ldots, f_k^N\right]$ are code phase and Doppler frequency, $\left[\rho_k^1, \ldots, \rho_k^N\right]$ and $\left[\dot{\rho}_k^1, \ldots, \dot{\rho}_k^N\right]$ are pseudoranges and pseudorange rates of 1th to $N$th satellite in the epoch $K$, respectively.

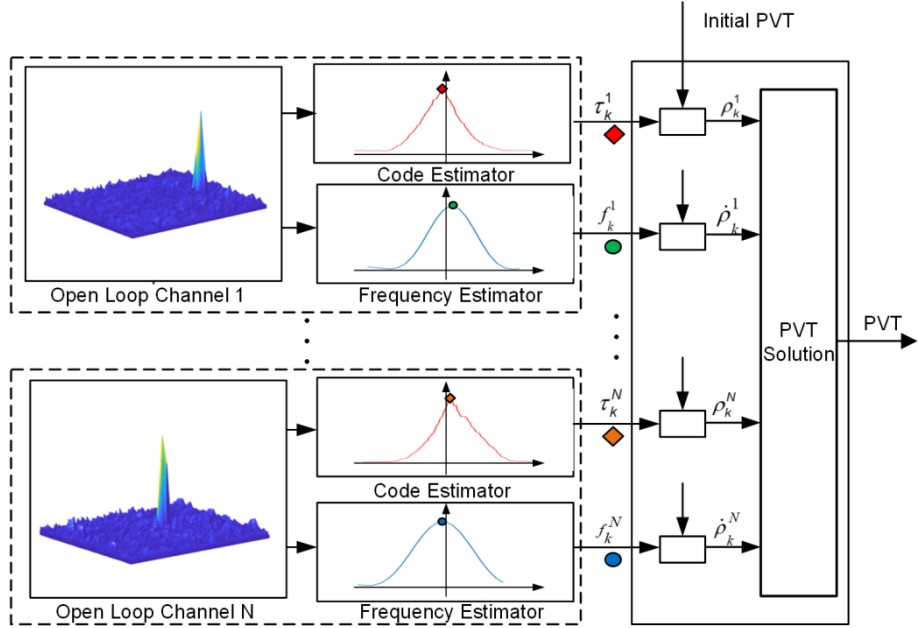

**Figure 1.** Diagram of conventional open loop tracking and position, velocity and time (PVT) solution.

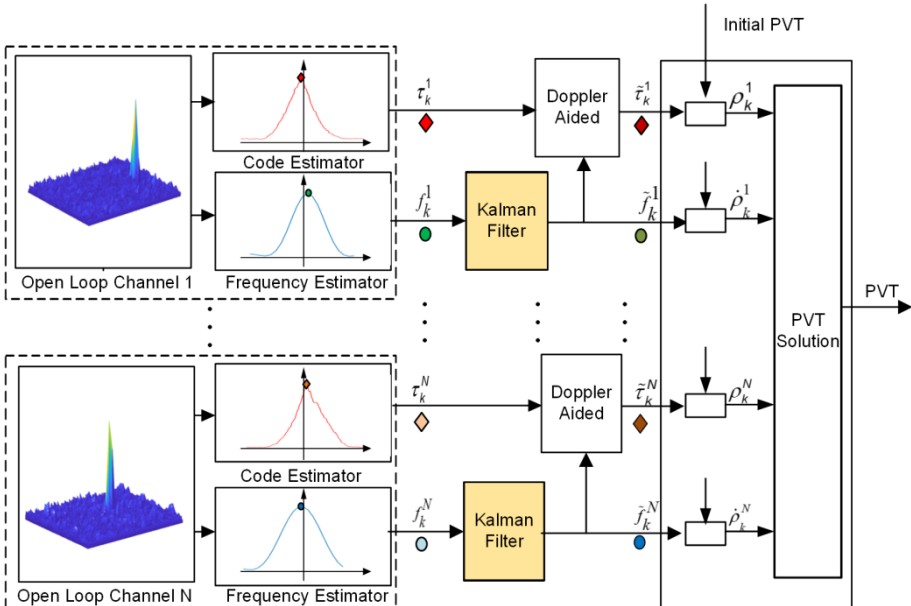

**Figure 2.** Diagram of open loop tracking with a Kalman filter and PVT solution.

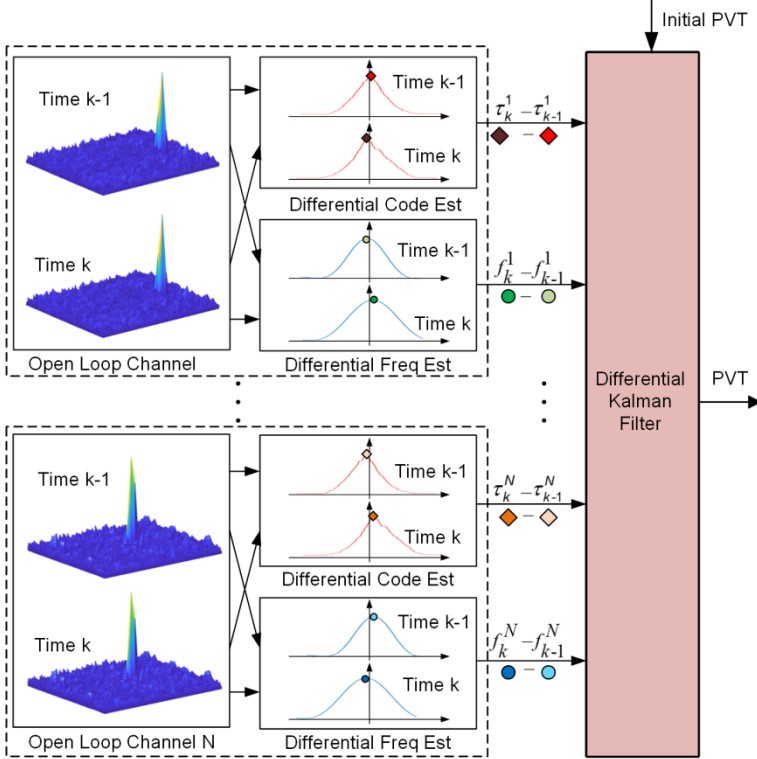

**Figure 3.** Diagram of open loop tracking with a differential Kalman filter.

### 2.1. State Model of the Proposed OL-DKF

In general, the state vector of a traditional receiver is shown as

$$X_k = [x_k, y_k, z_k, b_k, \dot{x}_k, \dot{y}_k, \dot{z}_k, d_k]^T \tag{1}$$

where $x_k$, $y_k$ and $z_k$ is the 3D position of the receiver in the Earth-Centered Earth-Fixed (ECEF) coordinate, $\dot{x}_k$, $\dot{y}_k$ and $\dot{z}_k$ is the 3D velocity of the receiver in the ECEF coordinate, $b_k$ is the receiver's clock bias and $d_k$ is the clock drift of receiver. The subscripts $k$ denote $k$-th epoch.

The state model of OL-DKF is based on the differential state between adjacent epochs. The state vector is the differential result of the traditional state vector, shown as:

$$\overline{X}_k = X_k - X_{k-1} = [\, \overline{x}_k, \overline{y}_k, \overline{z}_k, \overline{b}_k, \overline{\dot{x}}_k, \overline{\dot{y}}_k, \overline{\dot{z}}_k, \overline{d}_k]^T \tag{2}$$

where $(\bar{\ })_k$ means the differential value between epoch $k$ and $k-1$. $\overline{X}_k$ contains the differential position, differential clock bias, differential velocity and differential clock drift of the receiver.

The state transition equation is given by:

$$\overline{X}_k = \Phi\overline{X}_{k-1} + W_{k-1} \tag{3}$$

where $\Phi$ is the state transition matrix from epoch $k-1$ to $k$:

$$\Phi = \begin{bmatrix} 1 & 0 & 0 & 0 & T & 0 & 0 & 0 \\ 0 & 1 & 0 & 0 & 0 & T & 0 & 0 \\ 0 & 0 & 1 & 0 & 0 & 0 & T & 0 \\ 0 & 0 & 0 & 1 & 0 & 0 & 0 & T \\ 0 & 0 & 0 & 0 & 1 & 0 & 0 & 0 \\ 0 & 0 & 0 & 0 & 0 & 1 & 0 & 0 \\ 0 & 0 & 0 & 0 & 0 & 0 & 1 & 0 \\ 0 & 0 & 0 & 0 & 0 & 0 & 0 & 1 \end{bmatrix} \qquad (4)$$

$T$ is the time interval between epoch $k-1$ and $k$, $W$ is process noise which is the Gaussian white noise with zero mean, with covariance matrix $Q$, given by [18]

$$\begin{aligned} Q &= E[W_{k-1}W_{k-1}{}^T] \\ &= \begin{bmatrix} T^3\sigma_x^2/3 & 0 & 0 & 0 & T^2\sigma_x^2/2 & 0 & 0 & 0 \\ 0 & T^3\sigma_y^2/3 & 0 & 0 & 0 & T^2\sigma_y^2/2 & 0 & 0 \\ 0 & 0 & T^3\sigma_z^2/3 & 0 & 0 & 0 & T^2\sigma_z^2/2 & 0 \\ 0 & 0 & 0 & T\sigma_b^2 + T^3\sigma_d^2/3 & 0 & 0 & 0 & T^2\sigma_d^2/2 \\ T^2\sigma_x^2/2 & 0 & 0 & 0 & T\sigma_x^2 & 0 & 0 & 0 \\ 0 & T^2\sigma_y^2/2 & 0 & 0 & 0 & T\sigma_y^2 & 0 & 0 \\ 0 & 0 & T^2\sigma_z^2/2 & 0 & 0 & 0 & T\sigma_z^2 & 0 \\ 0 & 0 & 0 & T^2\sigma_d^2/2 & 0 & 0 & 0 & T\sigma_d^2 \end{bmatrix} \end{aligned} \qquad (5)$$

Here, $\sigma_x^2$, $\sigma_y^2$ and $\sigma_z^2$ are variances of process noise in the ECEF coordinate, $\sigma_b^2$ is the variance of the oscillator phase noise and $\sigma_d^2$ is the variance of the oscillator frequency noise.

### 2.2. Measurement Model of the Proposed OL-DKF

The measurement model of OL-DKF is based on the differential open loop tracking result between adjacent epochs, which is:

$$\overline{Z}_k = H_k \overline{X}_k + N_k \qquad (6)$$

where $\bar{z}_k = z_k - z_{k-1} = \left[\overline{\tau}_k^1, \cdots, \overline{\tau}_k^N, \overline{f}_k^1, \cdots, \overline{f}_k^N\right]^T$ and $z_k = \left[\tau_k^1, \cdots, \tau_k^N, f_k^1, \cdots, f_k^N\right]^T$. $\overline{\tau}_k^i$ and $\overline{f}_k^i$ are the differential code phase and differential Doppler frequency, which are equal to $\tau_k^i - \tau_{k-1}^i$ and $f_k^i - f_{k-1}^i$, the superscripts $i$ denote $i$th satellite. $N_k$ is the measurement noise, of which the covariance matrix is $R$. $H_k$ is transition matrix, given by:

$$H_k = \begin{bmatrix} \alpha_k^1 & \beta_k^1 & \gamma_k^1 & 1 & 0 & 0 & 0 & 0 \\ \cdots & \cdots & \cdots & \cdots & \cdots & \cdots & \cdots & \cdots \\ \alpha_k^N & \beta_k^N & \gamma_k^N & 1 & 0 & 0 & 0 & 0 \\ 0 & 0 & 0 & 0 & \alpha_k^1 & \beta_k^1 & \gamma_k^1 & 1 \\ \cdots & \cdots & \cdots & \cdots & \cdots & \cdots & \cdots & \cdots \\ 0 & 0 & 0 & 0 & \alpha_k^N & \beta_k^N & \gamma_k^N & 1 \end{bmatrix} \qquad (7)$$

where $\left[\alpha_k^i, \beta_k^i, \gamma_k^i\right]$ is the line-of-sight unit vector between the $i$th satellite and the receiver in ECEF coordinates.

The measurements, code phase $\tau_k^i$ and Doppler frequency $f_k^i$, have relationships with the state vector of the receiver $X_k$ and the state vector of $i$th satellite, shown as:

$$\tau_k^{i} = \frac{-f_{code}}{c}\left(\left[\alpha_k^i, \beta_k^i, \gamma_k^i\right]\begin{bmatrix} x_k - x_k^i \\ y_k - y_k^i \\ z_k - z_k^i \end{bmatrix} + (b_k - b_k^i)\right)$$

$$f_k^{i} = \frac{-f_{carr}}{c}\left(\left[\alpha_k^i, \beta_k^i, \gamma_k^i\right]\begin{bmatrix} \dot{x}_k - \dot{x}_k^i \\ \dot{y}_k - \dot{y}_k^i \\ \dot{z}_k - \dot{z}_k^i \end{bmatrix} + (d_k - d_k^i)\right) \tag{8}$$

$$\left[\alpha_k^i, \beta_k^i, \gamma_k^i\right] = \frac{\left[x_k - x_k^i, y_k - y_k^i, z_k - z_k^i\right]}{\sqrt{\left(x_k - x_k^i\right)^2 + \left(y_k - y_k^i\right)^2 + \left(z_k - z_k^i\right)^2}}$$

where $x_k^i$, $y_k^i$, $z_k^i$, $b_k^i$, $\dot{x}_k^i$, $\dot{y}_k^i$, $\dot{z}_k^i$, and $d_k^i$ are the position, clock bias, velocity and clock drift of $i$th satellite. $f_{code}$ is code rate, $f_{carr}$ is carrier frequency. $c$ is the speed of light.

## 3. Parameters Setting

In this chapter, performance analyses of OL-DKF, OL and OL-KF were carried out. OL-DKF combines the tracking filter and position procedures. Its outputs (PVT) are the position and the tracking result at same time. So, performance analyses among three methods mainly focus on the standard deviation of the 3D position and velocity. Considering the simplified assumptions of noise contributions (uncorrelated and identically distributed noise contributions), pseudoranges and pseudorange rates were employed to calculate PVT in OL and OL-KF so that we used the position dilution of precision (PDOP) factor to evaluate the position accuracy of OL and OL-KF. On the other hand, differential code phases and Doppler frequencies were employed to calculate PVT in OL-DKF so that Ricatti function was used to evaluate accuracy of OL-DKF.

### 3.1. Performance Analysis of OL-DKF

The performance analysis of the tracking loop was divided into the analysis on the input and output of the filter. In OL-DKF model, the input performance depends on the accuracies of the code phases and the Doppler frequencies from the OL tracking. The output performance depends on filter accuracy of the differential Kalman filter.

3.1.1. Input Performance of OL-DKF

Based on the detection probability, we could analyze the input performances of OL-DKF. In the OL, the integration result $V_k$ at epoch $k$ is obtained by coherent and non-coherent integration. The in-phase branch coherent integration result $I(k)$ and quadrature branch coherent integration result $Q(k)$ in epoch $k$ can be expressed as [19]:

$$\begin{cases} I(k) = A_k D(k) R_k(\Delta\tau_k)\sin c(\pi\Delta f_k T)\cos(\Delta\phi_k) + n_{I,k} \\ Q(k) = A_k D(k) R_k(\Delta\tau_k)\sin c(\pi\Delta f_k T)\sin(\Delta\phi_k) + n_{Q,k} \end{cases} \tag{9}$$

where $A_k$ is the signal amplitude, $D(k)$ is the navigation message, $R_k(\Delta\tau_k)$ is the code autocorrelation function, $\Delta\tau_k$ is the code phase error, $\Delta f_k$ is the Doppler frequency error and $\Delta\phi_k$ is the carrier phase error. $n_{I,k}$ and $n_{Q,k}$ are the uncorrelated Gaussian white noise.

The magnitude of the complex coherent integration $I(k) + jQ(k)$ without noise is:

$$V_k = \sqrt{I^2(k) + Q^2(k)} = A_k R_k(\Delta\tau_k)\left|\sin c(\pi\Delta f_k T_{coh})\right| \tag{10}$$

Assuming the variances of $n_{I,k}$ and $n_{Q,k}$ are $\sigma_n^2$, $V_k$ obeys Rayleigh distribution in the absence of satellite signal and Rice distribution in the presence of satellite signal. Based on the threshold and the probability density function of the Rice distribution $p(V)$, the detection probability $P_D$ is:

$$P_D = \int_{Th}^{\infty} p(V)dV \tag{11}$$

We could analyze the input accuracy of OL-DKF by the detection probabilities of the code phase and Doppler frequency in each search grid.

- Accuracy of the code phase measurement

The correlation peak between the local replica and incoming signal may occur in any of the search grids due to the presence of noise. Assuming the maximum amplitude occurs in *j*th ($j = 0, 1, \ldots, M$) search grid, and $M$ is the total number of the search grids, the detection probability of *j*th the search grid is $\int_0^{\infty} p(V, \Delta\tau(j))dV$. The probability that the amplitudes, except *j*th search grid, of all the search grids are less than the maximum amplitude is $\prod_{\substack{k=0 \\ k \neq j}}^{M} [1 - P_D(V, \Delta\tau(k))]$ [20]. The detection probability of the code error $\Delta\tau(j)$ in *j*th the search grid is

$$P_D(\Delta\tau(j)) = \int_0^{\infty} p(V, \Delta\tau(j)) \cdot \prod_{\substack{\Delta\tau(i) \in \tau_{range} \\ i \neq j}} [1 - P_D(V, \Delta\tau(i))]dV \tag{12}$$

where $p(V, \Delta\tau(j)) = V \cdot e^{-\frac{V^2 + a_\tau^2}{2}} I_0(a_\tau V)$, $a_\tau = \sqrt{(1 - |\Delta\tau(j)|)^2 \cdot T_{coh} \cdot C/N_0 \cdot 2}$ and $\tau_{range} = [-1, -1 + \tau_{step}, -1 + 2\tau_{step}, \ldots, 1]$ is the code search range, $\tau_{step}$ is code search step and $I_0(\cdot)$ is first kind zero-order modified Bessel's function.

Based on Equation (12), Figure 4 shows the detection probabilities of code phase error with a different carrier to noise ratio (C/N$_0$) when $T_{coh}$= 40 ms. The detection probability curves are approximately a normal distribution under high C/N$_0$ and tend to be uniformly distributed under low C/N$_0$ such as 16 dB-Hz. The standard deviation of code phase error $\sigma_\tau$ is:

$$\sigma_\tau = \sqrt{\sum_{\Delta\tau(j) \in \tau_{range}} (\Delta\tau(j))^2 \cdot P_D(\Delta\tau(j))} \tag{13}$$

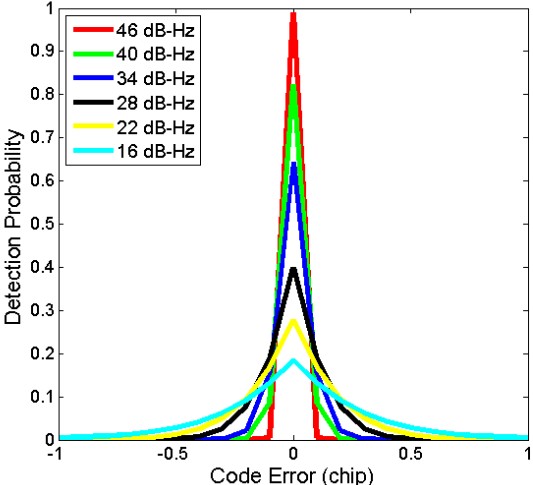

**Figure 4.** Detection probability of the code phase error.

Figure 5 depicts the standard deviation of code phase error based on Equation (13). The standard deviation of code phase error was lower when the search step was smaller or C/N0 was higher. So, the standard deviation of code phase error was related to C/N0 and the length of the code search step.

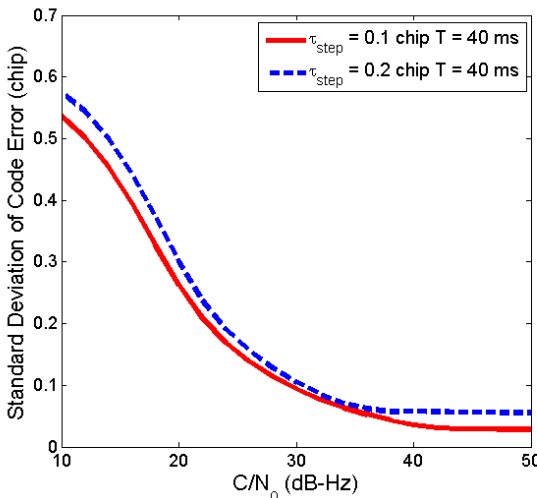

**Figure 5.** Code error of the open loop under thermal noise.

- Accuracy of the Doppler frequency measurement

The Doppler frequency detection probability can be similarly analyzed as above. When the maximum amplitude point corresponds to $j$th the search grid with the Doppler frequency error $\Delta f(j)$, the detection probability is:

$$P_D(\Delta f(j)) = \int_0^\infty p(V, \Delta f(j)) \cdot \prod_{\substack{\Delta f(i) \in frange \\ i \neq j}} [1 - P_D(V, \Delta f(i))] dV \tag{14}$$

where $p(V, \Delta f(j)) = V \cdot e^{-\frac{v^2 + a_f^2}{2}} I_0(a_f V)$, $a_f = \sqrt{\sin c^2(\Delta f(j) \cdot T_{coh}) \cdot T_{coh} \cdot C/N_0 \cdot 2}$ and $frange = [-f, -f + f_{step}, -f + 2f_{step}, \ldots, f]$ is the frequency search range and $f_{step}$ is carrier Doppler frequency search step.

Based on Equation (14), Figure 6 shows the detection probabilities of the Doppler frequency error with different C/N$_0$ and $T_{coh}$ = 40 ms. The curves are an approximately normal distribution under high C/N$_0$ and tend to be uniformly distributed under low C/N$_0$. The standard deviation of the Doppler frequency error $\sigma_f$ is shown as:

$$\sigma_f = \sqrt{\sum_{\Delta f(j) \in frange} (\Delta f(j))^2 \cdot P_D(\Delta f(j))} \tag{15}$$

According to Equations (14) and (15), the Doppler frequency error is related to the thermal noise and not related to the dynamic. Figure 7 shows the standard deviation of the Doppler frequency error when the frequency search step is 2.5 Hz and 5 Hz. The standard deviation of the Doppler frequency error was lower when the search step was smaller or C/N$_0$ was higher. It was related to C/N$_0$ and the length of frequency search step.

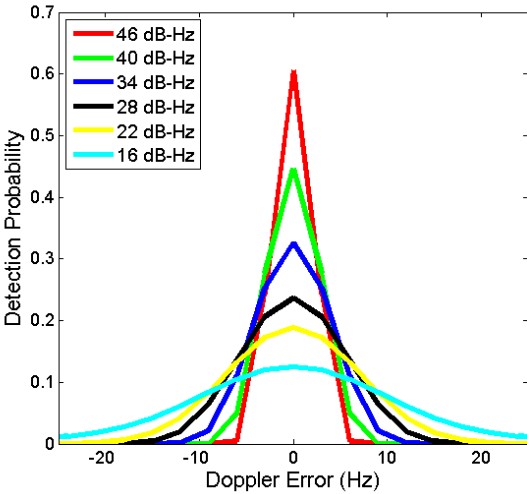

**Figure 6.** Detection probability of the Doppler frequency error.

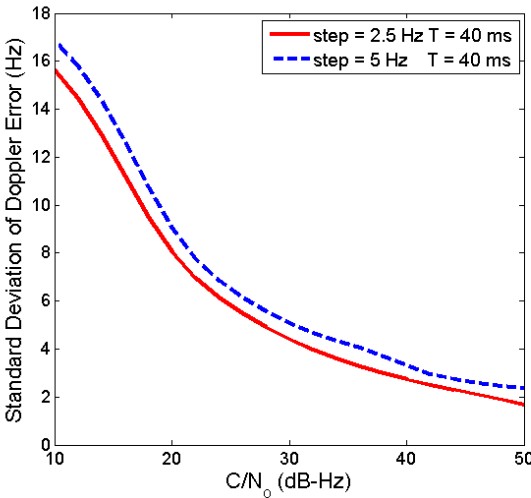

**Figure 7.** Doppler error of the open loop under thermal noise.

### 3.1.2. Output Performance of OL-DKF

The tracking accuracy of OL-DKF under the thermal noise can be calculated by the prior steady-state covariance matrix $P_{ss}^-$ via the Ricatti function:

$$\Phi P_{ss}^- \Phi^T - \Phi P_{ss}^- H^T (H P_{ss}^- H^T + R)^{-1} H P_{ss}^- \Phi^T + Q - P_{ss}^- = 0 \tag{16}$$

The posterior steady-state covariance matrix $P_{ss}^+$ contains processing noise and observation noise, which is given by:

$$P_{ss}^+ = (I - K_{ss}H)P_{ss}^-(I - K_{ss}H)^T + K_{ss}RK_{ss}^T \tag{17}$$

where the variances of measurement noise for each channel are $2\sigma_\tau^2$ and $2\sigma_f^2$, which can be derived from Equations (13) and (15). $K_{ss} = P_{ss}^- H^T (H P_{ss}^- H^T + R)^{-1}$ is the steady-state gain of OL-DKF. So, the estimate accuracies of position and velocity ($\sigma_{Pos}^{OL-DKF}$, $\sigma_{Vel}^{OL-DKF}$) are given by

$$\begin{aligned} \sigma_{Pos}^{OL-DKF} &= \sqrt{P_{ss}^+(1,1) + P_{ss}^+(2,2) + P_{ss}^+(3,3)} \\ \sigma_{Vel}^{OL-DKF} &= \sqrt{P_{ss}^+(5,5) + P_{ss}^+(6,6) + P_{ss}^+(7,7)} \end{aligned} \tag{18}$$

where $P_{ss}^+(i,j)$ represents the *i*th row and *j*th column element of $P_{ss}^+$.

### 3.2. Accuracy of OL Outputs

The tracking accuracy of the receiver will directly affect positioning accuracy. Usually, the position dilution of precision (PDOP) is used to describe the relationship between tracking accuracy and positioning accuracy, which is related to the geometric location of the receiver and satellites. The positioning and velocity accuracies of OL are shown as [19]:

$$
\begin{aligned}
\sigma_{Pos}^{OL} &= PDOP \times \sigma_\tau / f_{code} \times c \\
\sigma_{Vel}^{OL} &= PDOP \times \sigma_f / f_{carr} \times c
\end{aligned}
\tag{19}
$$

where $\sigma_{Pos}^{OL}$ is the standard deviation of 3D position of OL, $\sigma_{Vel}^{OL}$ is the standard deviation of 3D velocity of OL.

### 3.3. Accuracy of OL-KF Outputs

In the OL-KF architecture, the state vector contains carrier phase error and Doppler frequency error. The accuracy of Doppler frequency $\varepsilon_f$ can be obtained from the Kalman filter output. The accuracy of the original code phase is $\varepsilon_\tau$, which can be calculated based on the non-coherent early minus later power discriminator [14]. According to [21], the accuracy of smoothed pseudo-code $\varepsilon_{s-\tau}$ is shown as:

$$
\varepsilon_{s-\tau}^2 = \sqrt{\frac{\varepsilon_\tau^2}{L} + \frac{(c/f_{carr})^2 T_{coh}^2 \varepsilon_f^2}{4L^2}\left(\frac{2L(L-1)(2L-1)}{3} + (L-1)^2\right)}
\tag{20}
$$

where $L$ is the smoothing depth. Appendix A provides detailed deduction steps.

Similar to Equation (19), the position and velocity accuracies of OL-KF can also be derived as:

$$
\begin{aligned}
\sigma_{Pos}^{OL-KF} &= PDOP \times \varepsilon_{s-\tau} / f_{code} \times c \\
\sigma_{Vel}^{OL-KF} &= PDOP \times \varepsilon_f / f_{carr} \times c
\end{aligned}
\tag{21}
$$

where $\sigma_{Pos}^{OL-KF}$ is the standard deviation of 3D position of OL-KF and $\sigma_{Vel}^{OL-KF}$ is the standard deviation of 3D velocity of OL-KF.

## 4. Numerical Simulations and Comparisons Between OL, OL-KF and the Proposed OL-DKF

In this chapter, numerical simulations were conducted. The accuracy comparisons of position and velocity among three architectures were carried out based on the above chapter with simplified assumptions of noise contributions and no errors condition. There were two constellations used in the analysis. One constellation had 10 satellites, while the other had 6 satellites, shown in Figure 8. The 6 satellites were picked out from the 10 satellites. Performance comparisons under more comprehensive error conditions and bad geometry in the real city environment are shown in the next chapter.

### 4.1. Comparison of Velocity Accuracy

The velocity accuracies of OL-DKF, OL and OL-KF can be obtained from Equations (18), (19) and (21). Note that the process noise variance of the proposed OL-DKF in the three-dimensional directions $\sigma_x^2, \sigma_y^2$ and $\sigma_z^2$ are set to the same value $\sigma^2$. In the comparison between OL-KF and OL-DKF, an equivalent processing noise should be set. The process noise variance in OL-KF is $q_{los}^2$, which is caused by the acceleration along the line-of-sight (LOS) vector from the satellite to the receiver. The relationship between $\sigma^2$ and $q_{los}^2$ is $\sigma^2 = PDOP^2 \cdot q_{los}^2 / 3$.

Figures 9 and 10 show the velocity accuracies of the three architectures with different processing noises and satellite numbers. We could find that (a) velocity errors of three architectures were lower with increasing satellite number in view and C/$N_0$. (b) Velocity errors of OL-KF and OL-DKF were smaller than those of OL. (c) The proposed OL-DKF show the performance improvement compared to the OL-KF under the equivalent noise setting.

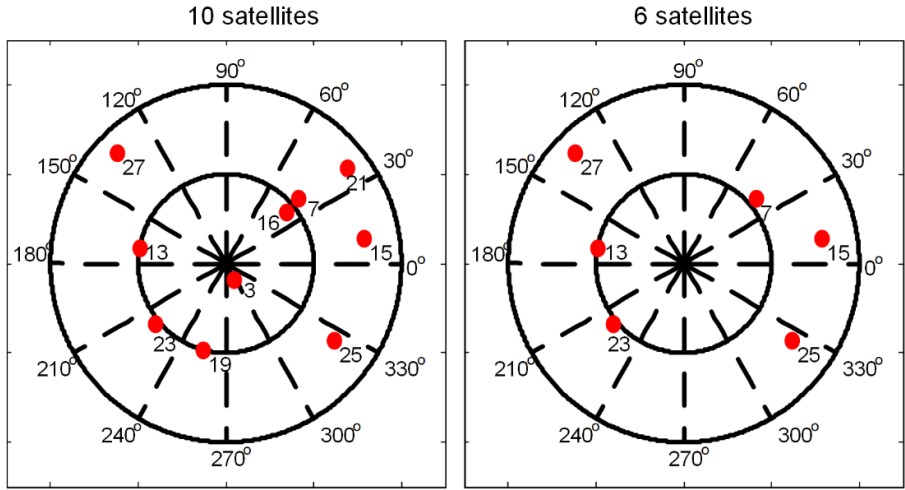

**Figure 8.** Satellite constellations.

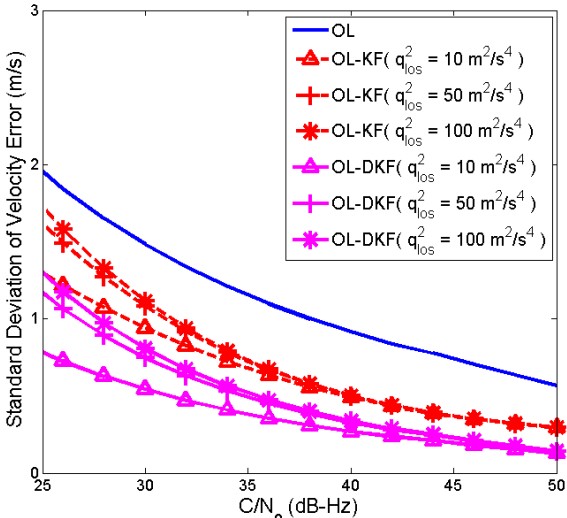

**Figure 9.** Standard deviation of the velocity error (10 sats).

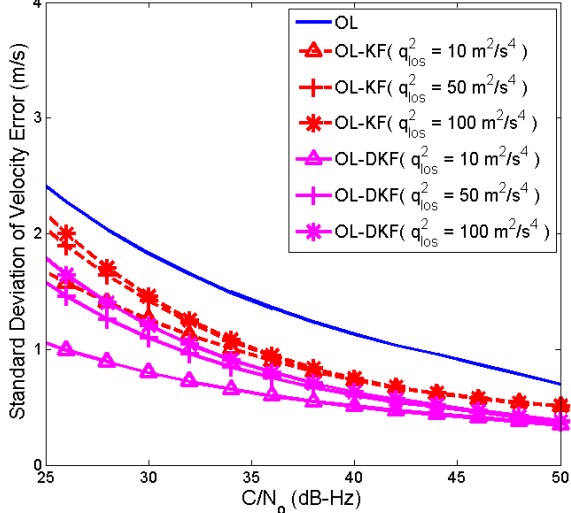

**Figure 10.** Standard deviation of the velocity error (6 sats).

### 4.2. Comparison of Position Accuracy

The position accuracies of OL-DKF, OL and OL-KF could be obtained from Equations (18), (19) and (21). Figures 11 and 12 show the position accuracies of the three architectures with different processing noise and satellite numbers when $L$ is set 15. The conclusion of the position accuracy analysis was similar to that of the velocity accuracy analysis.

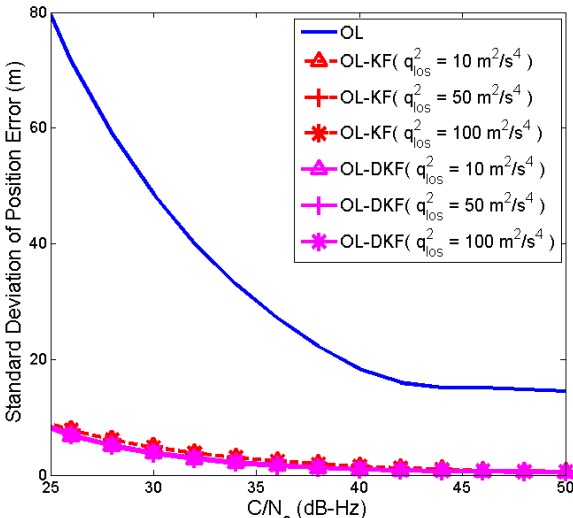

**Figure 11.** Standard deviation of the position error (10 sats).

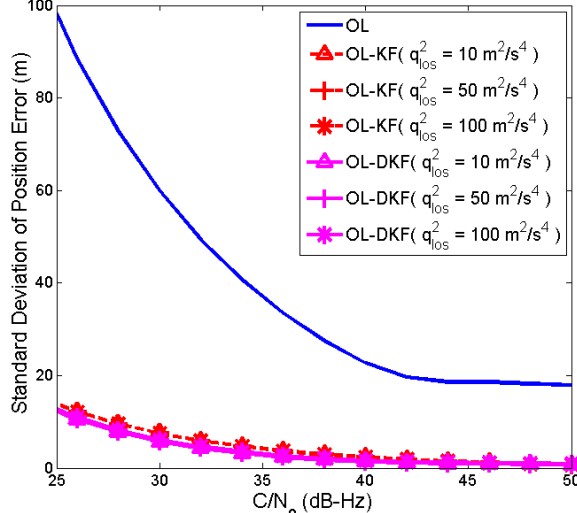

**Figure 12.** Standard deviation of the position error (6 sats).

## 5. Experiments and Results

Actual road tests were conducted to evaluate the performance of OL-DKF, OL-KF and OL in different urban test cases. Coherent time was set 40 ms. Road test cases included unblocked roads, roads blocked by light railway and roads in the city canyon. The GPS signal was collected by down conversion, filtering and analogue to digital sampling. Then, the digital intermediate frequency signal was processed by receivers. Figure 13 shows the actual route where experimental data were collected. The high precision inertial navigation system (IMU-FSAS) was used to record the vehicle trajectory, which is shown as the red line in Figure 13.

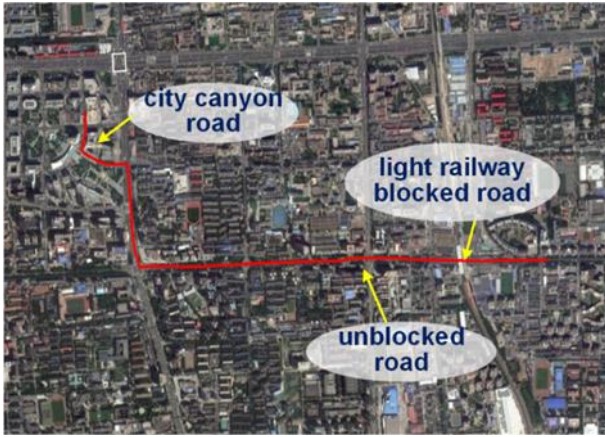

**Figure 13.** Road test routes.

### 5.1. First Test Case: Unblocked Roads

Figure 14 shows the 3D map of the section of the route where the road environment is open and SNR is high, but there are still a few high buildings that may cause interference to the receiver.

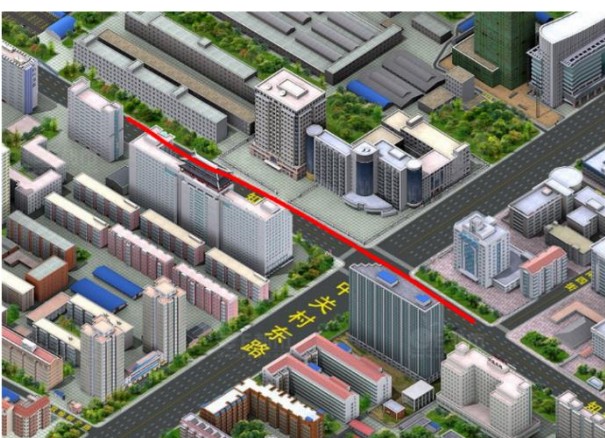

**Figure 14.** 3D map of the unblocked road.

The experimental results of this experiment are shown in Figures 15–19. It can be seen from Figure 15 that there was good satellite visibility in this section of the route. Figures 16 and 17 show that the position estimated by the proposed OL-DKF was less noisy and matched the data collected by the IMU-FSAS better than the OL and OL-KF. Figure 18 shows the probability distribution of position error where it can be observed that the position errors of OL-DKF were less than 2 m, while only 85% and 35% position errors of OL-KF and OL reached the same accuracy. Figure 19 shows three-dimensional errors in East-North-Up(ENU) coordinate. The standard deviations of horizontal errors of OL-DKF, OL-KF and OL were 0.4 m, 0.7 m and 1.5 m. The standard deviations of vertical errors of the above architectures were 9.1 m, 8.8 m and 12.1 m.

### 5.2. Second Test Case: Roads Blocked by Light Railway

Figure 20 shows the 3D road map of the section of the route where roads blocked by a light railway. Vehicle traversed the light railway from below. The yellow circle marks the light railway blockage area, which made the receiver fail to view satellite in a short period. OL-DKF took a reasonable prediction under this condition.

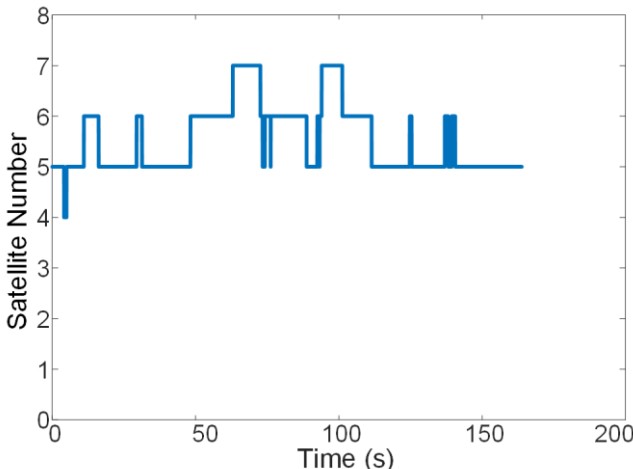

**Figure 15.** Satellite visibility of the unblocked road test.

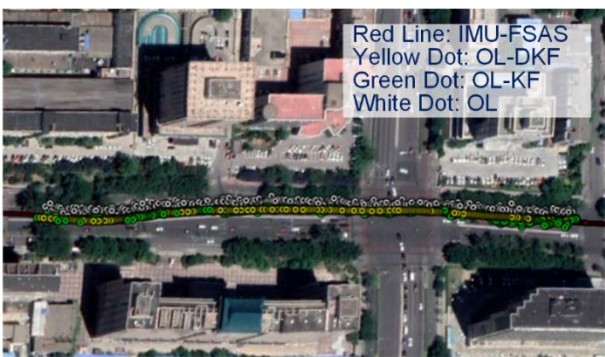

**Figure 16.** Trajectories of open loop with a differential Kalman filter (OL-DKF), open loop tracking aided by the Kalman filter (OL-KF), open loop tracking (OL) and IMU-FSAS of the unblocked road test.

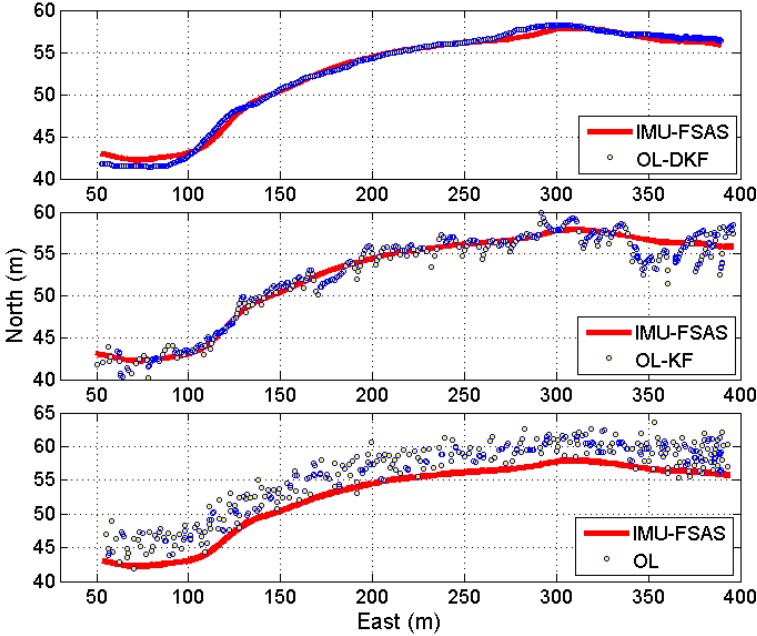

**Figure 17.** Position plot of OL-DKF, OL-KF and OL against IMU-FSAS of the unblocked road test.

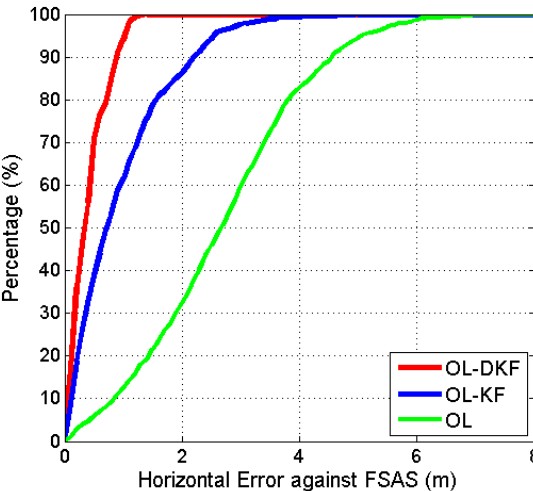

**Figure 18.** Position error probability of the unblocked road test.

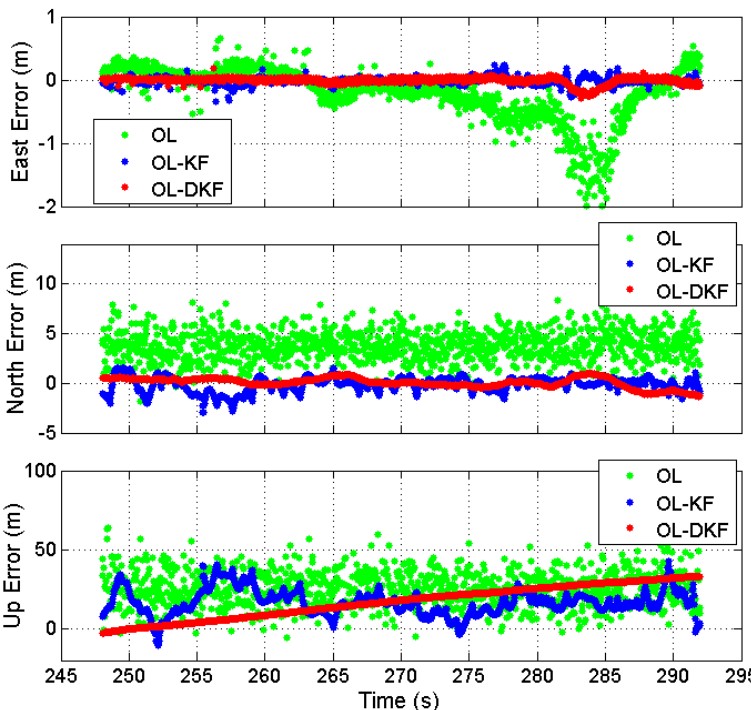

**Figure 19.** Three-dimensional errors of the unblocked road test.

The experimental results of this experiment are shown in Figures 21–25. It can be seen from Figure 21 that satellite numbers were stable, except for about 2 s when no satellite was in view. Figures 22 and 23 show that the position estimated by the proposed OL-DKF was less noisy and matched the data collected by the IMU-FSAS better than the open loop algorithm. The prediction of OL-DKF in the blocked area produced a satisfied result. Figure 24 shows the probability distribution of the position error and we can observe that 85% position errors of OL-DKF were less than 3 m, while 74% and 38% position errors of OL-KF and OL reached the same precision. Figure 25 shows three-dimensional errors in ENU coordinate. The standard deviations of horizontal errors of OL-DKF, OL-KF and OL were 0.9 m, 1.6 m and 2.1 m. The standard deviations of vertical errors of the above architectures were 10.7 m, 10.5 m and 22.2 m.

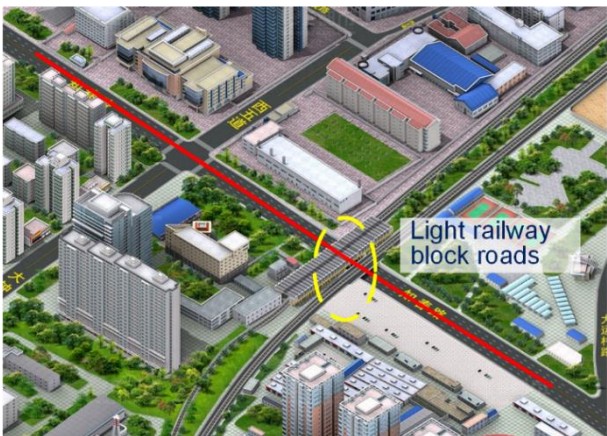

**Figure 20.** 3D map of roads blocked by a light railway.

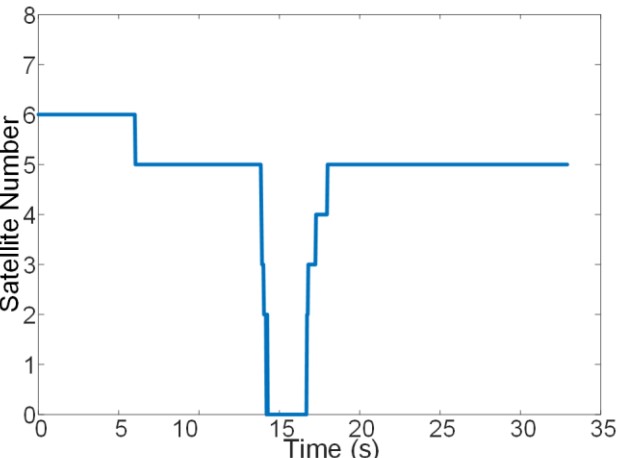

**Figure 21.** Satellite visibility of roads blocked by a light railway test.

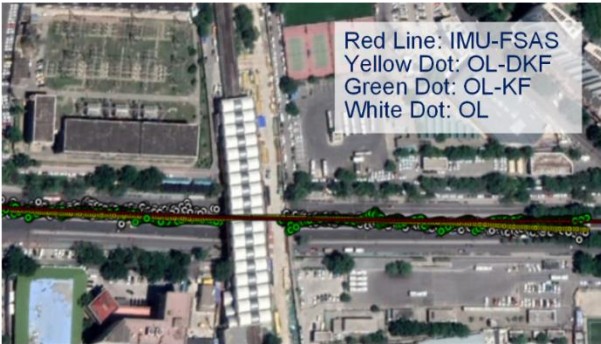

**Figure 22.** Trajectories of OL-DKF, OL-KF, OL and IMU-FSAS of roads blocked by a light railway.

*5.3. Third Test Case: Roads in City Canyon*

Figure 26 shows the 3D road map of roads in city canyon where many tall buildings along both sides of the roads affected the receiver considerably. The yellow circle marks the blocked area where visible satellites change rapidly, and even frequently drops below 4. Moreover, poor SNR and PDOP cause interference to the receiver. OL-DKF made reasonable predictions and constraints.

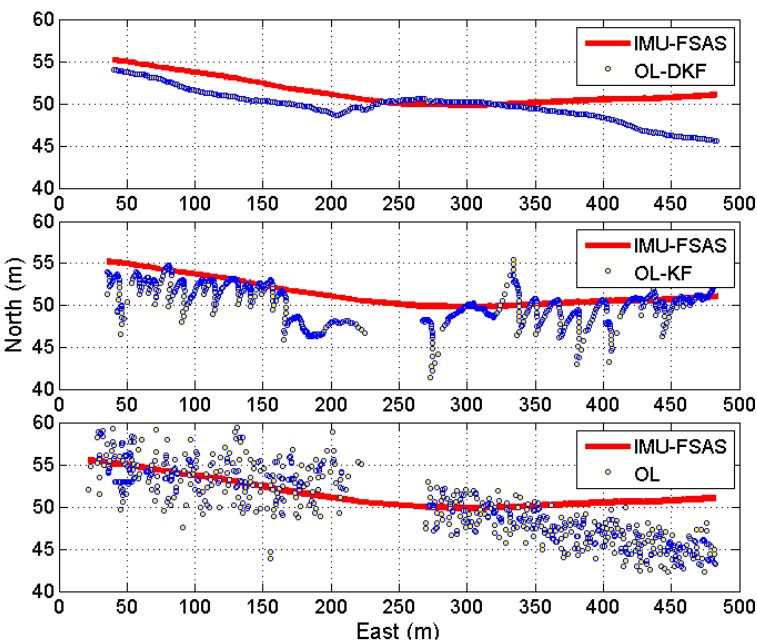

**Figure 23.** Position plot of OL-DKF, OL-KF and OL against IMU-FSAS of roads blocked by a light railway test.

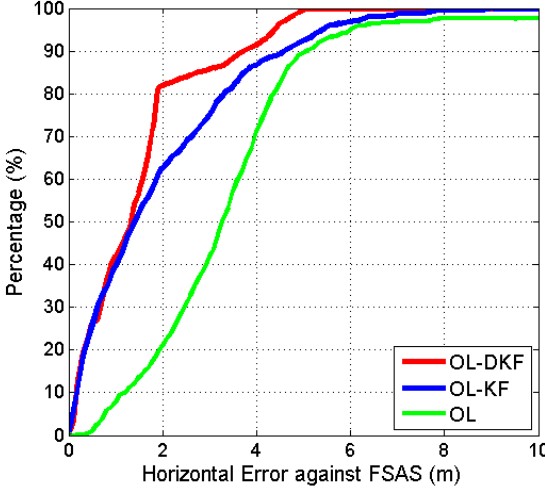

**Figure 24.** Position error probability of roads blocked by a light railway test.

The experimental results of this experiment are shown in Figures 27–31. It can be seen from Figure 27 that satellite numbers were unstable and less than four frequently. Figures 28 and 29 shows that the position estimated by the proposed OL-DKF was less noisy and much better than that by the open loop algorithm. Figure 30 is the probability distribution of the position error. It can be observed that 85% position errors of OL-DKF were within 10 m, while 51% and 45% of the position errors of OL-KF and OL reached the same precision. Figure 31 shows three-dimensional errors in ENU coordinate. The standard deviations of horizontal errors of OL-DKF, OL-KF and OL were 3.3 m, 5.1 m and 7.8 m. The standard deviations of vertical errors of the above architectures were 15.3 m, 20.5 m and 51.1 m.

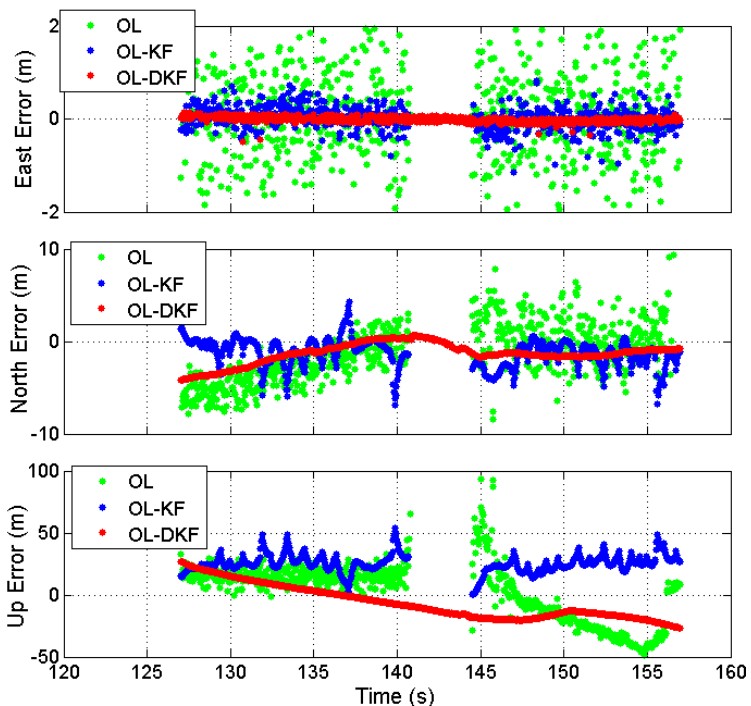

**Figure 25.** Three-dimensional errors of roads blocked by a light railway test.

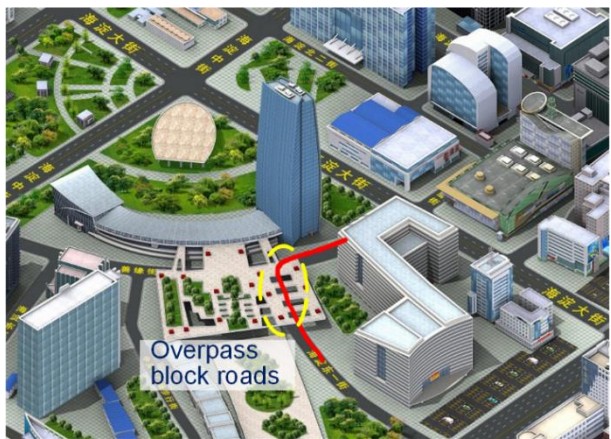

**Figure 26.** 3D map of roads in the city canyon test.

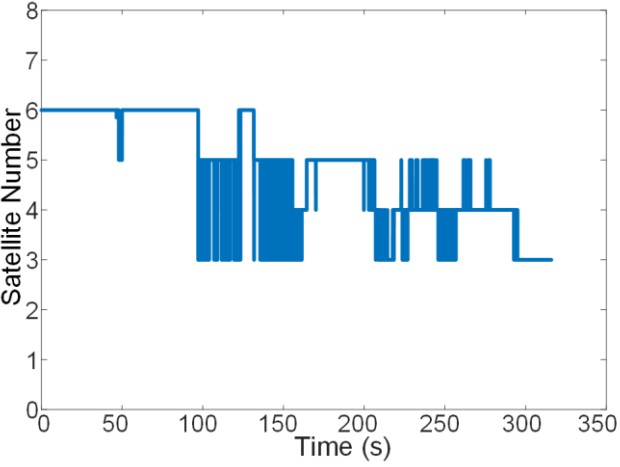

**Figure 27.** Satellite visibility of roads in the city canyon test.

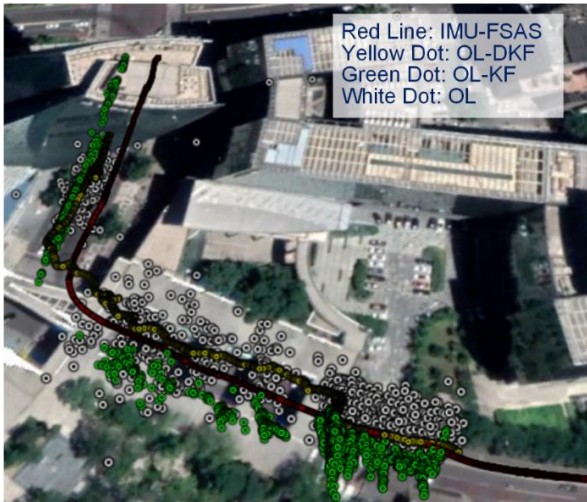

**Figure 28.** Trajectories of OL-DKF, OL-KF, OL and IMU-FSAS of roads in the city canyon test.

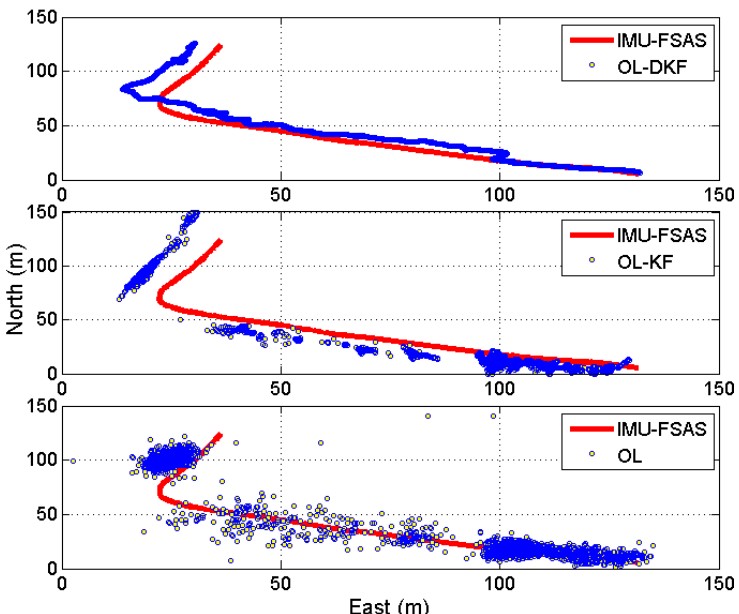

**Figure 29.** Position plot of OL-DKF, OL-KF and OL against IMU-FSAS of the roads in the city canyon test.

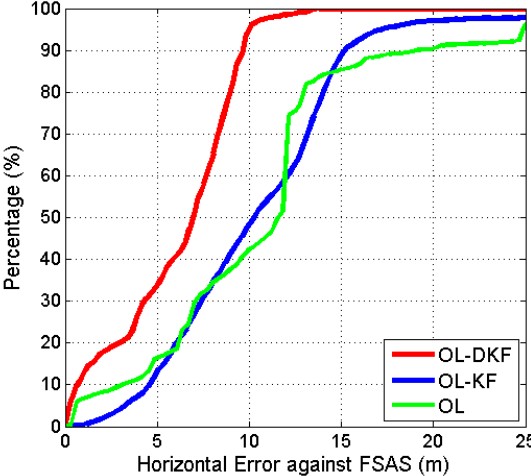

**Figure 30.** Position error probability of roads in the city canyon test.

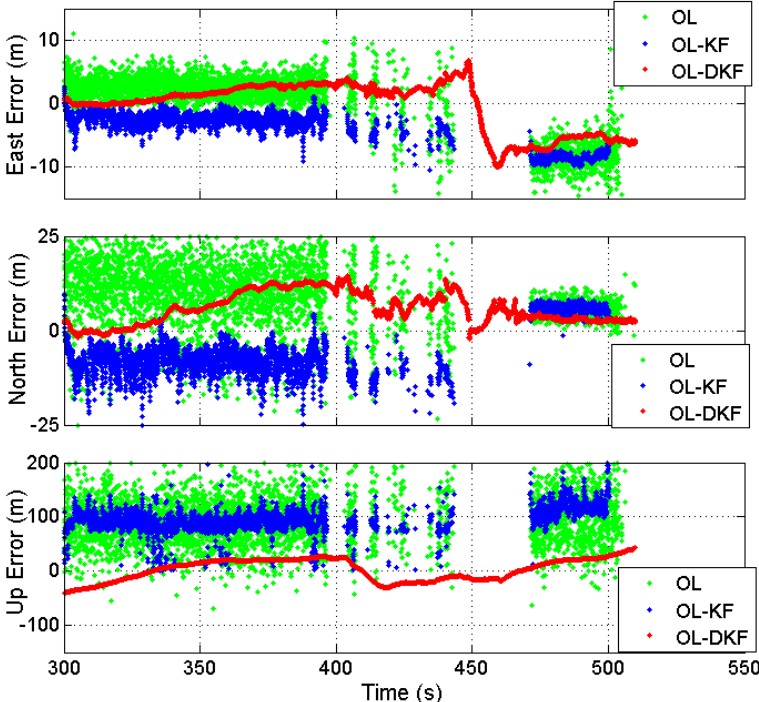

**Figure 31.** Three-dimensional errors of roads in the city canyon test.

## 6. Conclusions

We proposed an open loop algorithm based on the differential Kalman filter, with a detailed theoretical model. This architecture takes the differential values of open loop tracking results between adjacent epochs, unlike the traditional open loop, which uses code phase and frequency estimates as input. It reduces the influence of common errors in open loop tracking and obtains more accurate PVT solutions. We theoretically analyzed the input and output performance of the OL-DKF, and compared it with the traditional open loop and open loop with a Kalman filter. The performance of OL-DKF was better than the other algorithms because it utilized the motion relationship from the temporal domain and the geometry relationship from the spatial domain by a differential Kalman filter. Three typical road tests in the city canyon environment were carried out. The results of theory and experiments show that the proposed architecture had a better position and velocity accuracy. There were many existing techniques used in the challenged environment. Compared with inertial navigation system assist, the proposed method did not need additional sensor-assisted. Compared with open loop, the OL-DKF could achieve much accuracy improvements in the city canyon environment.

**Author Contributions:** Conceptualization, T.J.; Data curation, H.Y.; Funding acquisition, J.K.; Methodology, T.J. and H.Y.; Software, H.Y.; Supervision, H.Q.; Writing—original draft, H.Y.; Writing—review and editing, T.J. and K.-V.L.; All authors have read and agreed to the published version of the manuscript.

**Funding:** This research was funded by Shaanxi Key Laboratory of Integrated and Intelligent Navigation, grant number SKLIIN-20180111, SKLIIN-20190106.

**Conflicts of Interest:** The authors declare no conflicts of interest.

## Appendix A. Derivation of Accuracy of the OL-KF Output

In OL-KF architecture, the state vector contains the carrier phase error and Doppler frequency error. The state transition matrix $\Lambda = \begin{bmatrix} 1 & T \\ 0 & 1 \end{bmatrix}$ and the measurement matrix $H = \begin{bmatrix} 0 & 1 \end{bmatrix}$. The variance of

processing noise is $Q_a = \frac{q_{los}^2}{(c/f_{carr})^2} \begin{bmatrix} T^3/6 & T^2/2 \\ T^2/2 & T \end{bmatrix}$. The variance of measurement noise is the accuracy of the differential power frequency discriminator output [14], shown as:

$$R_f = \frac{\mu_0}{4 \cdot T_{coh} \cdot C/N_0}\left(1 + \frac{\mu_1}{T_{coh} \cdot C/N_0}\right) \tag{A1}$$

where $\theta = \pi f_{step} T_{coh}$, $\mu_0 = \frac{f_{step}^2[1-cos(2\theta)]}{(sinc(\theta)-cos(\theta))^3}$, $\mu_1 = \frac{1-sinc^2(2\theta)}{2sinc^2(\theta)[1-cos(2\theta)]}$.

The tracking accuracy of OL-KF can be indicated by the posterior covariance matrix $P^{OL-KF}$, which can be calculated by the Ricatti equation. The frequency estimation accuracy is given by:

$$\varepsilon_f = \sqrt{P^{OL-KF}(2,2)} \tag{A2}$$

The accuracy of the original code phase $\varepsilon_\tau$ of OL-KF can be calculated based on the non-coherent early minus later power discriminator, shown as [14]:

$$\varepsilon_\tau = \sqrt{\frac{\tau_{step}}{2 \cdot T_{coh} \cdot C/N_0}\left(1 + \frac{1}{(1-\tau_{step}) \cdot T_{coh} \cdot C/N_0}\right)} \tag{A3}$$

In Equation (20), the accuracy of smoothed pseudo-code $\varepsilon_{s-\tau}$ can be calculated by Equations (A2) and (A3).

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
