# Peer review of "Differential Kalman Filter Design for GNSS Open Loop Tracking"

_remotesensing, doi:10.3390/rs12050812_

Round 1

Reviewer 1 Report

The paper is well written and provides detailed explanation on the OL-DKF filter. It is also a good asset to provide empirical results with different test cases.

The presentation is good, even though there is a general problem with the units in the images. Units should be between parenthesis and with a space between the parenthesis and the axis label text. The space is not left in the image axis labels, and others do not have the parenthesis before the unit. Figures 5, 7, 9, 10, 11, 12, 15, 17, 18, 19, 21, 23, 24, 25, 26, 28, 29 and 31 have to be redone with the labels fixed.

More specific comments:

1 In line 37, "Kalman Filter" should be changed to "KF", as you have defined the acronym in line 35.

2 In line 63, you state "an initial PVT solution is obtained from conventional receiver". What is the minimum accuracy needed for this initial solution? As OL-DKF is designed to work in challenging environments, the user could do a cold start in a challenging environment, thus the initial solution will be degraded.

3 In line 116 the dot between the parenthesis shall be removed, as the dot refers to the velocity, while the intention here is to show that the the dash over the character means "difference between epochs".

4 In line 125, would it be possible to write the Q matrix in the same way as in equation 4? This will make much easier to match with the transisiton matrix.

5 In line 125, where are the Q values coming from? There is no explanation or reference for the reason of using these values.

6 In line 190 you use 10 ms of coherent time. Why did you choose this value? (Later in the experimental results you use 40ms).

7 At the beginning of section you select a constellation of 10 satellites and a subset of these one with 6 satellites. Why and how do choose these constellation? Is it based on a real constellation or it has been invented to emphasize the performance of OL-DKF?

8 On line 268 you introduce the term "acceleration noise variance of line-of-sight". Could you add a little explanation of what is this value, how do you compute it and the tipycal range values (or a reference with the tipycal range values)? You use it later in figures 10, 11 and 12 with values of 10, 50 and 100 m^2/s^4, which it is not clear why these values are used.

9 In line 283, there is the phrase "when L is set to 15". What is L? And why 15? There is no other reference to this L in the paper.

10 In section 5, you use real data collected in urban canyon and you show the number of visible satellites, but it is not stated anywhere what constellation(s) are you using. Is it GPS, Galileo, GLONASS, BeiDou, QZSS or a combination of these? Also, it would be interesting if you could write differentiate between the type of orbits of the satellites used (MEO, IGSO, GEO), as the IGSO are mainly designed to be in higher elevation in certain areas (which is really useful for urban canyon areas).

11 In line 292, you use a coherent time of 40 ms. Why this time? In section 3 you used 10 ms. Furthermore, this integration time cannot create problems if a bit of the navigation message flips? In GPS, a bit in the navigation message lasts 20 ms, therefore with 40 ms you could have a bit change.

12 In section 5, you collect satellite data but there is no mention to what measurement you are using (is it C1C)? Please you RINEX observation notation. This info is important, as if you use a Pilot signal, the problem stated in the previous question does not occur.

14 It is possible to change the scale in figures 17 and 23? In lines 308 and 332 you state that errors are around 3 metres, but the scale is of 100 metres, which is too big to see errors of few metres clearly.

15 Legend of figures 16, 17, 22, 23, 27 and 28 have "OL-KF" missing. That is, instead of "OL-DKF and OL-KF" it should be "OL-DKF, OL-KF and OL".

16 Legend of figures 17, 23 and 28, the "and IMU-FSAS" should be replaced by "versus IMU-FSAS" or "against IMU-FSAS" or equivalent.

17 Legend of figure 19 is written as "Figure 199" instead of "Figure 19".

18 In figure 25, in the Up component, the legend of "OL-DKF" is wrong, it is written as "OL-KF".

19 In figure 25, in the Up component, is there any explanation for the continuous drift downwards of OL-DFK?

20 Figure 31 should be figure 30 (or is there a figure missing?).

21 In line 369, the phrase "with a theoretical model in details" should be reworded to "with a detailed theoretical model".

Author Response

Response for the reviewer’s comments of the paper

Dear Reviewer:

We would like to thank you very much for your comments, which are immensely useful for us to improve our paper. In this letter, we summarize the modifications on the manuscript, in order to respond to your comments. We will be glad to respond to any further questions and comments that you may have.

Sincerely,

Tian Jin, Heliang Yuan, Keck-Voon Ling, Honglei Qin and Jianrong Kang

The paper is well written and provides detailed explanation on the OL-DKF filter. It is also a good asset to provide empirical results with different test cases.

The presentation is good, even though there is a general problem with the units in the images. Units should be between parenthesis and with a space between the parenthesis and the axis label text. The space is not left in the image axis labels, and others do not have the parenthesis before the unit. Figures 5, 7, 9, 10, 11, 12, 15, 17, 18, 19, 21, 23, 24, 25, 26, 28, 29 and 31 have to be redone with the labels fixed.

Answer: Thanks for your advice. We modified the figures. Please see the revised manuscript for the detail modification.

More specific comments:

1、 In line 37, "Kalman Filter" should be changed to "KF", as you have defined the acronym in line 35.

Answer: Thanks for your advice. We modified "Kalman Filter" to "KF". The detail modification is as follow:

LINE 38: Psiaki [6] analyzed the performances of the Kalman filter KF in a weak signal.

2、 In line 63, you state "an initial PVT solution is obtained from conventional receiver". What is the minimum accuracy needed for this initial solution? As OL-DKF is designed to work in challenging environments, the user could do a cold start in a challenging environment, thus the initial solution will be degraded.

Answer: In the OL-DKF initialization, the observation information of the initial PVT was calculated by the conventional receiver. It is taken as a part of the measurement of OL-DKF at t = 0. Then, the initial position error will be gradually corrected by the filter in the OL-DKF. Commonly, the initial error can be corrected when the code phase error is within one chip. In the challenging environment, the error of the receiver may reach dozens of meters, but it is still within the range of one chip. Thus, the initial error can be corrected by subsequent filtering.

3、In line 116 the dot between the parenthesis shall be removed, as the dot refers to the velocity, while the intention here is to show that the dash over the character means "difference between epochs".

Answer: Thanks for your advice. We removed dot between the parenthesis. The detail modification is as follow:(or see the upload docx)

LINE 118: where  means the……

4、In line 125, would it be possible to write the Q matrix in the same way as in equation 4? This will make much easier to match with the transisiton matrix.

Answer: Thanks for your advice. We wrote the Q matrix in the same way as in equation 4. The detail modification is as follow:

LINE 130:  see the upload docx

5、In line 125, where are the Q values coming from? There is no explanation or reference for the reason of using these values.

Answer: we are sorry to make you confused. Q is process noise covariance matrix. We added explanation and the reference in the revised manuscript and the detail modification is as follow:

LINE 125:  is process noise which is the Gaussian white noise with zero mean, with covariance matrix , given by[18]

Reference:

  1. Hsu L T, Jan S S, Groves P D, et al. Multipath mitigation and NLOS detection using vector tracking in urban environments[J]. Gps Solutions, 2015, 19(2): 249-262.

6、In line 190 you use 10 ms of coherent time. Why did you choose this value? (Later in the experimental results you use 40ms).

Answer: In the performance analysis, our purpose is to compare the performance of OL, OL-KF and OL-DKF with the same coherent time. So, there is no limitation for the coherent time. According to your suggestion, we modified the coherent time to 40ms. See the revised manuscript for the detail modification.

7、At the beginning of section you select a constellation of 10 satellites and a subset of these one with 6 satellites. Why and how do choose these constellation? Is it based on a real constellation or it has been invented to emphasize the performance of OL-DKF?

Answer: The constellation is based on a real GPS satellite constellation. The purpose is to analyze and compare the OL-DKF with others.

8、On line 268 you introduce the term "acceleration noise variance of line-of-sight". Could you add a little explanation of what is this value, how do you compute it and the typical range values (or a reference with the typical range values)? You use it later in figures 10, 11 and 12 with values of 10, 50 and 100 m^2/s^4, which it is not clear why these values are used.

Answer: We are sorry to make you confused. Actually, we want to indicate the LOS dynamics as driving forces as the system process noise. However, our expression maybe makes readers confused. We modify it to “The process noise variance in OL-KF is q2los, which is caused by the acceleration along the line-of-sight (LOS) vector from the satellite to the receiver”. Considering the different dynamics, we set 10, 50 and 100 m2/s4 correspond to the low, medium and high dynamic level. In the article “Lashley M, Bevly D M, Hung J Y. Performance analysis of vector tracking algorithms for weak GPS signals in high dynamics[J]. IEEE Journal of selected topics in signal processing, 2009, 3(4): 661-673.”, the authors also set the process noise covariance on the three-dimensional coordinates as 2, 12, 50, 125 m2/s4, which is the same magnitude as ours.

The detail modification is as follow:

LINE 275:The acceleration noise variance of line-of-sight (LOS) vector to the satellite The process noise variance in OL-KF is, which is caused by the acceleration along the line-of-sight (LOS) vector from the satellite to the receiver.

9、In line 283, there is the phrase "when L is set to 15". What is L? And why 15? There is no other reference to this L in the paper.

Answer: L is the smoothing length of the Doppler smoothing pseudocode, which is illustrated in line 257. In the chapter of "simulation and experiment" of Lin's article, the smoothing length is set to 10. Considering the balance of convergence rate and tracking accuracy, we set the smoothing length to 15.

10、In section 5, you use real data collected in urban canyon and you show the number of visible satellites, but it is not stated anywhere what constellation(s) are you using. Is it GPS, Galileo, GLONASS, BeiDou, QZSS or a combination of these? Also, it would be interesting if you could write differentiate between the type of orbits of the satellites used (MEO, IGSO, GEO), as the IGSO are mainly designed to be in higher elevation in certain areas (which is really useful for urban canyon areas).

Answer: In section 5, the satellites signal we collected is GPS signal. The positioning divergence using different orbital types of the satellites is a very good research direction. We will conduct more research in the future work.

The detail modification is as follow:

LINE 305: The GPS signal was collected by down conversion, filtering and analogue to digital sampling. Then, the digital intermediate frequency signal was processed by receivers.

11、In line 292, you use a coherent time of 40 ms. Why this time? In section 3 you used 10 ms. Furthermore, this integration time cannot create problems if a bit of the navigation message flips? In GPS, a bit in the navigation message lasts 20 ms, therefore with 40 ms you could have a bit change.

Answer: Actually, the tracking performance is better with the longer integration time. The 40ms setting is a trade-off setting considering the improvement of sensitivity and complexity. We have modified the coherent time form 10ms to 40ms. The 40ms setting also needs to consider the message flips. So we use exhaustive method to detect the message flips. The method can be found in article” Luo P, Petovello M G. Collaborative Tracking of Weak GPS Signals Using an Open-loop Structure[J]. Proceedings of the ION ITM, 2011”. On the other hand, the coherent time will not be affected by the message flips when using pilot signal in future satellite signal.

12、In section 5, you collect satellite data but there is no mention to what measurement you are using (is it C1C)? Please you RINEX observation notation. This info is important, as if you use a Pilot signal, the problem stated in the previous question does not occur.

Answer: We are sorry to make you confused. The GPS signal was collected by down conversion, filtering and analogue to digital sampling. Then, the digital intermediate frequency signal was processed by receivers. The flow chart for data processing is shown on below:

The detail modification is as follow:

LINE 305: The GPS signal was collected by down conversion, filtering and analogue to digital sampling. Then, the digital intermediate frequency signal was processed by receivers.

14、It is possible to change the scale in figures 17 and 23? In lines 308 and 332 you state that errors are around 3 metres, but the scale is of 100 metres, which is too big to see errors of few metres clearly.

Answer: Thanks for your advice. The detail modification is as follow:(or see the upload docx)

Figure 17. Position plot of OL-DKF, OL-KF, OL against IMU-FSAS of unblocked road test

Figure 23. Position plot of OL-DKF, OL-KF, OL against IMU-FSAS of roads blocked by light railway test

15、Legend of figures 16, 17, 22, 23, 27 and 28 have "OL-KF" missing. That is, instead of "OL-DKF and OL-KF" it should be "OL-DKF, OL-KF and OL".

Answer: Thanks for your advice. We added the “OL-KF” in legend of figures 16, 17, 22, 23, 27 and 28. Please see the revised manuscript for the detail modification.

16、Legend of figures 17, 23 and 28, the "and IMU-FSAS" should be replaced by "versus IMU-FSAS" or "against IMU-FSAS" or equivalent.

Answer: Thanks for your advice. We changed the "and IMU-FSAS" to "against IMU-FSAS". Please see the revised manuscript for the detail modification.

17、Legend of figure 19 is written as "Figure 199" instead of "Figure 19".

Answer: Thanks for your advice. We modified the above error.

18、In figure 25, in the Up component, the legend of "OL-DKF" is wrong, it is written as "OL-KF".

Answer: Thanks for your advice. We modified the above error.

19、In figure 25, in the Up component, is there any explanation for the continuous drift downwards of OL-DFK?

Answer: We think that the continuous drift in the up component in OL-DKF is caused by the inaccuracy of the ionospheric correction model of the receiver in the up direction. In figure 25, we can see that the up components of OL and OL-KF also have different intensities of drift. Secondly, the drift of up direction of OL-DKF is slowly approaching to that of OL with filtering process.

20、Figure 31 should be figure 30 (or is there a figure missing?).

Answer: Thanks for your advice. There is just wrong figure number rather than figure missing. We have adjusted some figure numbers. Please see the revised manuscript for the detail modification.

21、In line 369, the phrase "with a theoretical model in details" should be reworded to "with a detailed theoretical model".

Answer: Thanks for your advice. We modified the phrase "with a theoretical model in details"  to "with a detailed theoretical model"

The detail modification is as follow:

LINE 404: with a detailed theoretical model in details.

Reviewer 2 Report

The method and tests are well described. I have only a few remarks:

In your experiments there is a rate/trend in the Up error for OL-DKF. Any potential reason ?

What happened to Up error in case of OL (figure 31). Why they are splitted? How they are calculated ?

Please add some values in section 5. It will be good to see the std or other factors.

Author Response

Response for the reviewer’s comments of the paper

Dear Reviewer:                                                

We would like to thank you very much for your comments, which are immensely useful for us to improve our paper. In this letter, we summarize the modifications on the manuscript, in order to respond to your comments. We will be glad to respond to any further questions and comments that you may have.

Sincerely,

Tian Jin, Heliang Yuan, Keck-Voon Ling, Honglei Qin and Jianrong Kang

The method and tests are well described. I have only a few remarks:

1、In your experiments there is a rate/trend in the Up error for OL-DKF. Any potential reason ?

Answer: We think that the continuous drift in the up component in OL-DKF is caused by the inaccuracy of the ionospheric correction model of the receiver in the up direction. In figures, we can see that the up components of OL and OL-KF also have different intensities of drift. Secondly, the drift of up direction of OL-DKF is slowly approaching to that of OL with filtering process.

2、What happened to Up error in case of OL (figure 31). Why they are splitted? How they are calculated ?

Answer: we are sorry to make you confused. Actually, when we plotted the figure, we made a mistake about the positive and negative of some data. We replotted the figure on below:(or see the upload docx )

Figure 31. Three-dimensional errors of roads in city canyon test

3、Please add some values in section 5. It will be good to see the std or other factors.

Answer: Thanks for your advice. We added some tables to illustrate std of three-dimensional errors.

The detail modification is as follow:

LINE 323: The standard deviations of horizontal errors of OL-DKF, OL-KF and OL are 0.45m, 0.77m and 1.55m. And the standard deviations of vertical errors of the above architectures are 9.1m, 8.8m and 12.1m.

LINE 354: The standard deviations of horizontal errors of OL-DKF, OL-KF and OL are 0.9m, 1.6m and 2.1m. And the standard deviations of vertical errors of the above architectures are 10.7m, 10.5m and 22.2m.

LINE 384: The standard deviations of horizontal errors of OL-DKF, OL-KF and OL are 3.3m, 5.1m and 7.8m. And the standard deviations of vertical errors of the above architectures are 15.3m, 20.5m and 51.1m.

This manuscript is a resubmission of an earlier submission. The following is a list of the peer review reports and author responses from that submission.